# Herbicidal Effects and Cellular Targets of Aqueous Extracts from Young *Eucalyptus globulus* Labill. Leaves

**DOI:** 10.3390/plants10061159

**Published:** 2021-06-07

**Authors:** Mafalda Pinto, Cristiano Soares, Maria Martins, Bruno Sousa, Inês Valente, Ruth Pereira, Fernanda Fidalgo

**Affiliations:** 1GreenUPorto-Sustainable Agrifood Production Research Centre, Departamento de Biologia, Faculdade de Ciências, Universidade do Porto, Rua do Campo Alegre, 4169-007 Porto, Portugal; up201506457@fc.up.pt (M.P.); c.fsoares@fc.up.pt (C.S.); up201306375@fc.up.pt (M.M.); bruno.filipe@fc.up.pt (B.S.); ruth.pereira@fc.up.pt (R.P.); 2REQUIMTE, LAQV, Departamento de Química e Bioquímica, Faculdade de Ciências, Universidade do Porto, Rua do Campo Alegre, 4169-007 Porto, Portugal; ines.valente@fc.up.pt; 3REQUIMTE, LAQV, ICBAS, Instituto de Ciências Biomédicas Abel Salazar, Universidade do Porto, Rua Jorge Viterbo Ferreira, 228, 4050-313 Porto, Portugal

**Keywords:** sustainable agriculture, allelopathy, bioherbicide, phytotoxic effects, weed management

## Abstract

*Eucalyptus globulus* Labill. is a widespread exotic species that contributes to the formation of fire-prone environments, a great concern under climate change conditions. Therefore, sustainable practices to help locals managing eucalyptus stands are needed. In this perspective, harnessing eucalyptus’ specialized metabolism as a source of allelochemicals can be a promising approach for weed control. Thus, the main goals of this work were to evaluate the herbicidal potential of post-fire regenerated *E. globulus* leaves against *Portulaca oleracea* L. and to unravel the physiological mechanisms behind this phytotoxic action. For this, aqueous extracts of fresh (FLE; 617 g FW L^−1^) or oven-dried leaves (DLE; 250 g DW L^−1^) were foliar-sprayed at different dilutions in purslane seedlings. After five weeks, results revealed that DLE at the highest dose detained the greatest herbicidal activity, affecting purslane growth and cellular viability. Moreover, biochemical data pointed towards an overproduction of reactive oxygen species, causing harsh oxidative damage in roots, where the upregulation of important cellular players, like sugars, amino acids, and proline, was not able to reestablish redox homeostasis. Overall, this study proved that dried leaves from young *E. globulus* had potent herbicidal properties against *P. oleracea* and can represent a feasible strategy for weed management.

## 1. Introduction

Weeds are the main elements responsible for crop yield and quality losses, since both weeds and crops compete for the same resources, such as light, water, minerals, and even space [1]. For this reason, modern agriculture is highly dependent on the use of synthetic herbicides, which can cost up to USD 7 billion annually [2]. However, although these herbicides are effective in eliminating weeds and avoiding major economic losses, their continued application has resulted in herbicide resistance acquisition by weed species [2] and, also, in the contamination of several environmental matrices, such as air, soil, and water [3,4,5], where they can negatively affect non-target organisms, including crops [6,7]. For this reason, new, eco-friendly, and economic solutions to combat weeds need to be urgently developed.

Some organisms, like plants, bacteria, and fungi, have the ability to produce specialized metabolites “that influence the growth and development of agricultural and biological systems” [8]. These organic compounds, commonly known as allelochemicals, can belong to different chemical families [8] and be released into the environment, impairing multiple physiologic and metabolic processes of target organisms [9]. This is the case of *Eucalyptus globulus* Labill. subsp. *globulus*, usually referred to as common eucalyptus or Tasmanian blue gum, a hardwood tree native from southeast Australia, now widely distributed throughout the world mainly due to its favorable wood characteristics for the paper industries [10]. Although flowers, fruits, bark, and roots have allelochemicals in their composition, the recognized *E. globulus* allelopathic activity is mainly attributed to its leaves [11]. Hence, the allelopathic properties of *E. globulus* leaves can be used in a targeted way to suppress the growth of undesirable plant species, such as weeds [12].

Thus far, multiple studies have been performed aiming at finding effective synthetic herbicide alternatives, taking advantage of the allelopathic properties of eucalyptus leaves [13,14,15]. Most of these studies evaluated the herbicidal potential of concentrated essential oils from *Eucalyptus* species [14,16]. However, leaf essential oils are obtained mainly by hydrodistillation procedures, which require expensive and specialized equipment [17], making them dependent on industrial transformation. As such, aqueous extracts prepared with leaves from eucalyptus trees, as well as from other plant species, can constitute affordable alternatives, allowing the obtention of not only essential oils but also polyphenolic compounds from the leaves, while ensuring the easiness and reproducibility of the procedure. Nonetheless, there are still some gaps regarding the use of eucalyptus-based products such as biocides.

Up to now, most works used extracts from *E. citriodora* [13,18,19,20] and *E. camaldulensis* [20,21,22]. Nevertheless, due to the favorable properties of its wood for the pulpwood industries, *E. globulus* is the most recurrent exotic species in world temperate regions, such as Portugal and Spain [10,23]. In this sense, the biocidal properties of this species have recently attracted more attention, with few studies evaluating the potential of concentrated essential oils [14,16], leachates [24,25], and aqueous extracts [11,12,26,27]. However, concerning aqueous extracts, most of the studies conducted so far evaluated the inhibitory effects of *E. globulus* extracts using *in vitro* bioassays, concluding that they are capable of effectively impairing seed germination and seedling growth of *Solanum nigrum* L. [26], *S. melongena* L. [27], and *Hordeum vulgare* L. [11], leaving much to clarify about the post-emergent biocidal properties of *E. globulus* aqueous extracts.

Moreover, when preparing new bioherbicides from plants, it is critical to pay attention to the processing of the leaves prior to the extraction procedure. For instance, when studying the phytotoxic potential of extracts from cultivated cardoon (*Cynara cardunculus* L.), Scavo et al. [28] reported that dried leaves could constitute a better option in terms of efficacy and cost management ratio than fresh or lyophilized ones. However, there is no consensus in the literature on this matter, including for *E. globulus* aqueous extracts. While some authors use fresh leaves in their studies [12,25], others prepare the aqueous extracts from dried leaves [11,26,27]. Indeed, only one study [24] has compared the herbicidal activity of leachates prepared with fresh and dried *E. globulus* leaves against *Cyperus rotundus* L. and *Cynodon dactylon* L., concluding that the one prepared with fresh leaves detained the greatest biocidal potential [24].

In addition, although some studies omit the age of the eucalyptus leaves utilized, only the mature leaves, collected from adult *E. globulus* trees, were used to prepare the aqueous extracts or leachates tested until now [12,24,29]. Therefore, the herbicidal potential of aqueous extracts prepared with young eucalyptus leaves constitutes a gap in knowledge and needs further investigation.

Due to their smaller heights, young eucalyptus trees or young resprouts tend to be more susceptible to herbivory than mature trees and to have higher amounts of defensive specialized metabolites than those of adult trees [30]. The distinctive morphology between young and adult *E. globulus* leaves also points to possible differences in their chemical composition: while leaves from young trees are blue-gray and have a thin and broad shape, mature leaves are typically dark green, thick, and narrow [31]. Hence, the allelopathic activity of *E. globulus*, aside from being related to its genetics and physiological state [9], may closely depend on its age.

Furthermore, the use of the leaf biomass from young eucalyptus trees for biocide production may represent a sustainable approach to control eucalyptus populations, outside plantations, especially in a climate change scenario within the Iberian Peninsula, a region with a high and growing incidence of wildfires [32]. Due to their fire-adaptation traits, when affected by a wildfire, *E. globulus* trees have a great ability to produce new shoots and thus, lead to the re-establishment of a vast leaf area, mainly by resprouting from dormant buds located in above-ground parts of the trees [33,34,35], allowing for a rapid dispersion of the species, even over areas previously occupied by native trees [35].

With this in mind, this study aims to evaluate the post-emergent herbicidal potential of the leaves (fresh and dried ones) of young *E. globulus* trees, resprouted after a wildfire, as well as to compare its effectiveness with synthetic herbicides, against a model weed species, *Portulaca oleracea* L. (common purslane). Being an annual weed species, usually found in fields of several cereals and horticultural crops such as maize (*Zea mays* L.), tomato (*Solanum lycopersicum* L.), sunflower (*Helianthus annuus* L.), and rice (*Oryza sativa* L.), *P. oleracea* is today considered one of the most recurrent and widespread weeds, with reports suggesting that global warming may even increase its adaptability and growth [36]. In order to get an insight into the extracts’ possible mode-of-action, a targeted biochemical assessment of which biomolecules and cellular processes could have been affected by the biocide was performed. 

## 2. Results

### 2.1. Evaluation of the Biocidal Potential of the Aqueous Extracts in Purslane Plants—Optimization of the Most Effective Dose and Extract (Fresh Leaf Extract or Dried Leaf Extract)

After foliar-applying the fresh leaf extract (FLE) and the dried leaf extract (DLE) to *P. oleracea* seedlings, the percentage of viable plants, macroscopic phytotoxicity symptoms, growth-related parameters, and cell death were analyzed. Glyphosate (GLY) was used as a positive control (CTL).

#### 2.1.1. Percentage of Viable Plants over Time

The variation of the percentage of viable plants that were treated with FLE or DLE can be observed in Figure 1a,b, respectively (Appendix A). Concerning FLE, the percentage of viable plants suffered a slight reduction of about 17% in the last week of the experiment, when the plants were treated with FLE at 100% (v/v), compared to the CTL (Figure 1a). Regarding DLE, a decrease in the percentage of viable plants treated with the highest concentration was observed from the third week until the end of the assay, following the same trend of GLY, although not so evident (Figure 1b). In the last week of the assay, a reduction of 35% in the percentage of viable plants was recorded for the treatment to which DLE at 100% (v/v) was applied to purslane plants, in comparison to the CTL plants. For the same week, the single application of GLY induced a 75% decrease in this parameter, compared to the CTL.

#### 2.1.2. Macroscopic Phytotoxicity Symptoms, Biometric Parameters, and Biomass Production

As shown in Figure 2, the inhibitory effects were more pronounced with the application of DLE, especially at 100% (v/v), which, in turn, were similar to those caused by GLY (positive CTL). Furthermore, the plants treated with the highest concentration of DLE, only developed one new leaf pair, with a lower leaf area, and a wilting appearance. Additionally, the application of FLE and DLE at any of the tested concentrations induced leaf chlorosis.

Both aqueous extracts affected the growth of purslane plants, inducing different effects depending on the applied concentration (Figure 2 and Table 1). Indeed, as the results of the two-way ANOVA suggest, a significant interaction between both factors (type of extract and applied concentration) was found for all biometric parameters analyzed (Appendix A), suggesting that the biocidal potential depended not only on the dilution, but also on the nature of the used leaves (fresh vs. dried). The foliar spraying with low doses (12.5, 25, and 50% (v/v)) of FLE enhanced shoot elongation by about 15%, 20%, and 29%, respectively, compared to the CTL. At higher concentrations (75 and 100% (v/v)), the shoot length was only affected (decreased ca. 26%) when the extract at 100% (v/v) was applied. Likewise, the shoot biomass followed a similar trend: FLE at 25% (v/v) and 50% (v/v) induced an increase of 34% and 58%, respectively, while a reduction by about 32% was measured with the application of FLE at 100% (v/v) (Table 1). When it comes to roots, while the length was not significantly altered by the application of FLE at any of the tested concentrations, the fresh weight was significantly enhanced when *P. oleracea* plants were treated with FLE at 50% (v/v) and 75% (v/v), increasing 39% and 43%, respectively (Table 1).

Regarding the treatment with DLE, the shoot length was enhanced by about 22% with the application of the extract at 50% (v/v) and was impaired, approximately by 36%, when the DLE was applied at 100% (v/v) (Table 1). With respect to the biomass of the aerial part, a significant reduction (ca. 52%) in response to DLE was found with the application of the extract at 75% (v/v), decreasing more than 70% with the spraying of the extract at 100% (v/v), compared to CTL (Table 1). This reduction in shoot biomass induced by the application of DLE at 100% (v/v) was statistically equivalent to the one obtained with the application of GLY, which decreased this parameter by about 88%, compared to CTL (Table 1). In what concerns root length, DLE severely affected this endpoint when it was applied at the two highest concentrations (75% (v/v)—decrease of 50%; 100% (v/v)—decrease of 65%, compared to the CTL) (Table 1). Regarding root biomass, the foliar spraying of purslane plants with DLE at 75% (v/v) and 100% (v/v) caused a decrease in root biomass by 56% in both situations, in comparison to the CTL. The maximum inhibition rates for all evaluated endpoints (root length and biomass) were recorded for GLY-treated plants (Table 1). The statistical results of ANOVA analyses for shoot and root length and biomass are presented in Appendix A.

#### 2.1.3. Cell Viability

The histochemical detection of cell viability of *P. oleracea* plants treated with different concentrations of FLE and DLE, as well as with GLY, is represented in Figure 3 and Figure 4. In general, and compared to CTL, FLE only induced cell death when sprayed at 100% (v/v), where a pattern of cell death similar to the one obtained with GLY was found (Figure 3). In contrast, the application of DLE decreased cell viability in all tested concentrations, in a dose-dependent manner, with the highest concentration showing a similar cell death pattern to that of GLY (Figure 4).

### 2.2. Assessment of the Phytotoxicity Induced by the Application of the Optimized Aqueous Extract on Purslane Plants

#### 2.2.1. Redox Status

As can be seen in Table 2, malondialdehyde (MDA) levels in purslane shoots were not significantly altered when the two concentrations (75 and 100% (v/v)) of DLE were applied. In contrast, the content of MDA considerably increased, approximately by 89%, in the roots of plants treated with the highest concentration of the eucalyptus extract (Table 2), compared to the untreated situation.

Concerning hydrogen peroxide (H_2_O_2_), the levels of this reactive oxygen species (ROS) suffered a 3- and 5-fold increase with the spraying of the highest concentration (100% (v/v)) of DLE in shoots and roots, respectively, compared to CTL (Table 2). The ANOVA statistical results of MDA and H_2_O_2_ levels are denoted in Appendix A.

#### 2.2.2. Total Sugars, Proteins, Free Amino Acids, and Proline Content

Regarding the total sugar levels, the treatment with the highest concentration of DLE increased this parameter by 3 times in the shoots and roots of purslane plants, compared to the CTL situation (Table 2 and Appendix A).

Total soluble protein levels were substantially increased (about 89%) in the aerial part of purslane plants, with the application of DLE at 100% (v/v) (Table 2 and Appendix A). In contrast, in the roots, the same concentration of DLE considerably decreased this parameter by about 50%, relative to the CTL situation (Table 2 and Appendix A).

As can be seen in Table 2 and Appendix A, total free amino acid levels in *P. oleracea* shoots were reduced (26% over the CTL) only when the 75% (v/v) DLE was sprayed. Regarding the roots, the application of the highest concentration of the extract significantly increased the levels of total free amino acids by approximately 59% (Table 2 and Appendix A). With regard to proline levels, there was a significant increase of about 37% in the shoots of purslane plants treated with DLE at the highest concentration, compared to the CTL situation (Table 2 and Appendix A).

#### 2.2.3. Levels of Photosynthetic Pigments

As shown in Table 2, total chlorophylls (a and b) were significantly impaired with the foliar spraying of the extract at 75% (v/v), decreasing the levels of these pigments by about 28% compared to the untreated situation (Appendix A). However, while the treatment with the highest concentration of DLE did not alter total chlorophyll levels, carotenoid production increased 19% when this dose was applied to purslane plants, compared to CTL (Table 2 and Appendix A).

#### 2.2.4. Nitrogen (N) Metabolism

The impact of DLE application on N nutrition of purslane plants was evaluated through the quantification of two enzymes belonging to N metabolism, nitrate reductase (NR; EC 1.6.6.1), and glutamine synthetase (GS; EC 6.3.1.2), in the aerial part and in the roots of *P. oleracea* plants. As shown in Table 2 and Appendix A, NR activity significantly increased in the shoots of purslane plants treated with 75% (v/v) DLE, with an increase of 58% over the CTL, while in the roots, the NR activity did not change with the treatments (Table 2). With regard to GS, the results revealed that although the activity of this enzyme was not altered in the aerial part of the plants (Table 2 and Appendix A), this enzyme was positively affected (67%) in the roots by the highest applied concentration of DLE, in relation to the CTL situation (Table 2 and Appendix A).

## 3. Discussion

The herbicidal potential of *E. globulus* leaf extracts, which has been explored by some authors [11,12,24,25,26,27] in the last years, is strictly linked to the presence of considerable amounts of volatile compounds, mainly from the monoterpene and sesquiterpene classes, and of polyphenols, such as ellagitannins, flavonols, and phenolic acids [14,37,38,39]. However, most of the studies conducted so far have focused on the pre-emergent activity of *E. globulus* extracts, evaluating their effects only on seed germination and seedling growth. From all these studies, only Puig et al. [12] evaluated the biocidal potential of an aqueous extract prepared with fresh leaves in a post-emergent context. In that study, the authors used leaves from mature trees to prepare the extract and concluded that it possessed a biocidal action when applied daily, by foliar spraying or watering, for 3 weeks on *Lactuca sativa* L., an agronomically relevant species, not evaluating its activity towards weeds. Furthermore, the type of *E. globulus* leaves preferred in the preparation of the aqueous extracts is still not unanimous: while some reports take advantage of the allelopathic properties of fresh leaves to produce the extracts [12,25], others use exclusively the dried ones [11,26,27]. To the best of our knowledge, the only study dedicated to the comparison of the herbicidal potential between fresh and dried leaf extracts used biomass from mature *E. globulus* trees [24]. In this sense, the present study is the first to evaluate the post-emergent biocidal activity of aqueous extracts prepared with fresh or dried leaves obtained from young *E. globulus* trees, which are typically richer in specialized metabolites, such as those with strong allelopathic activities. For that, 7-days-old purslane seedlings were foliar sprayed, twice a week, with several dilutions of FLE (617 g_fresh weight_ L^−1^) and DLE (250 g_dry weight_ L^−1^).

### 3.1. DLE Shows a High Biocidal Potential, Impairing Weed Growth and Compromising Cell Viability

The application of FLE at low concentrations, especially at 50% (v/v), presented curious fertilizing properties, enhancing plant growth, namely shoot elongation and biomass production. As expected, the treatment with 50% (v/v) FLE did not alter the percentage of viable plants over the five weeks of the experiment and did not induce an increase of cell death in purslane plants. Altogether, these results show that, at least up to 50% (v/v), the FLE did not have toxic effects on purslane plants. In fact, the allelopathic activity of eucalyptus leaves against other plants is only effective when the allelochemicals are present at certain concentrations [1], which were not achieved with the use of the diluted formulation.

In contrast, the foliar-treatment with the DLE at concentrations up to 50% (v/v) did not induce relevant alterations in *P. oleracea* plants. Indeed, even though a significant increase in the shoot length was observed upon the treatment with the DLE at 50% (v/v), all the other endpoints remained unchanged. This may be due to the drying process of the leaves, which, on one hand, may have induced the loss of volatile compounds and, on the other, may have caused the accumulation of greater amounts of compounds with allelopathic properties, such as hydrolysable tannins [40]. In fact, the chromatographic analyses revealed that the DLE presented higher amounts of galloyltannins and a greater variety of ellagitannins, two classes of compounds belonging to the group of hydrolysable tannins, than the other extract (Pinto et al., in prep). The FLE, in turn, had a more abundant content of volatile compounds, mainly monoterpenes (Pinto et al., in prep).

In opposition, at higher doses (75 and especially 100% (v/v)), FLE has been shown to negatively affect plant growth, inducing a pattern of cell death similar to that obtained for the treatment with GLY, the most widely used synthetic herbicide [41]. Although the application of 100% (v/v) FLE only induced a significant reduction in the percentage of viable plants after 5 weeks, the results, altogether, evidence that the FLE at 100% (v/v) had an effective herbicidal activity against purslane plants, but with a slower action compared to the synthetic herbicide. In fact, GLY presented a strong inhibitory effect of purslane growth right in the initial stages of seedling development, significantly decreasing the percentage of viable plants after the first two weeks.

Accordingly, the treatment of *P. oleracea* plants with the highest concentration of DLE (100% (v/v)) caused the greatest reduction in all tested parameters compared to FLE at its maximum concentration. As a matter of fact, DLE at 250 g_dry weight_ L^−1^ was the only treatment capable of effectively reducing the percentage of viable plants in the time frame of the experiment, which evidences that DLE detained a strong and effective biocidal activity. However, these findings are not in agreement with the works of El-Rokiek and Eid [19] and Babu and Kandasamy [24], whose results showed that the aqueous extracts or leachates, respectively, prepared with fresh leaves had a more powerful herbicidal activity against the tested weed species than the ones obtained from dried leaves. These distinct outcomes might be attributed to the different procedures employed for the preparation of the tested biocides. While the extracts optimized in the present study resulted from a hot-water extraction, the aqueous extracts and leachates used in the studies of El-Rokiek and Eid [19] and Babu and Kandasamy [24] were prepared by incubating fresh and dried leaves from *E. citriodora* and *E. globulus*, respectively, in cold water for 48 h. This difference in the temperature chosen for the extraction process may have influenced the type and quantity of compounds with allelopathic activity extracted from the fresh and dried leaves, which then may have been translated into distinct herbicidal activities.

Despite the observed differences between FLE and DLE, the application of both eucalyptus-based herbicides induced leaf chlorosis. The explanation for this observation may reside in the mode-of-action of these biocides. Plant-based herbicides, like eucalyptus extracts, can affect weed growth by disrupting nutrient uptake, photosynthetic processes, and membrane permeability [42]. Briefly, the allelochemicals present in plant herbicides can alter the structure and function of cell membranes, decreasing the absorption of macro and micronutrients, such as magnesium (Mg), potassium (K), and iron (Fe) [42]. Therefore, the observed chlorosis in purslane leaves treated with FLE and DLE may result in the reduction of Mg, K, and Fe intracellular levels, whose presence is essential for the normal function of chloroplasts [43]. In fact, while Mg is present at the active center of the chlorophyll molecule, Fe composes many cytochromes and nonheme iron proteins involved in the photosynthetic metabolism, and K is crucial for maintaining cell electroneutrality and participates in a series of photosynthetic reactions [43]. Accordingly, a reduction in the levels of total chlorophylls was detected in the plants treated with DLE at 75% (v/v). Although not statistically significant, probably due to the large decrease in the percentage of viable plants, the content of chlorophylls a and b in plants treated with the most effective concentration of DLE, presented a downward trend. Indeed, an analogous study in which an aqueous extract prepared with fresh *E. globulus* leaves was applied to lettuce plants reported that the application of the extract induced a reduction in the total chlorophyll content, which was accompanied by a decline in chlorophyll a fluorescence [12]. These results allowed the authors to conclude that the light-independent reactions of photosynthesis were directly affected by the application of the extract [12].

Moreover, DLE at 100% (v/v) caused foliar wilting, which is usually associated to a loss of membrane integrity [42]. Indeed, a study conducted with the purpose of assessing the herbicidal potential of *E. citriodora* oil on *Phalaris minor* Retz. reported the occurrence of wilting and chlorotic leaves in the treated plants, which was the result of a high degree of ion leakage induced by the biocide application, causing the loss of membrane permeability and integrity [44]. In accordance, the intracellular H_2_O_2_ levels were greatly increased in the shoots of the purslane plants treated with the highest concentration of DLE.

Overall, the herbicidal activity shown by DLE at 100% (v/v) was comparable to that of GLY, although with an apparently slower action, but much faster and effective than that of FLE. In fact, the time lag to have an effect from DLE and FLE applications may depend mainly on the time needed to reach an effective dose, with the successive applications, rather than on the mode-of-action of the bioherbicides. Indeed, the reduction of the growth-related parameters and the pattern of cell death of the plants treated with DLE at 100% (v/v) were identical to the ones obtained when GLY was applied. In addition, the onset of the herbicidal effects of DLE was registered only one week apart from that of GLY (Figure 1). Thus, from all the tested extracts and concentrations, it was possible to conclude that the DLE at 100% (v/v) had the greatest herbicidal activity against *P. oleracea* plants.

### 3.2. The Biocidal Potential of DLE Is Linked to an Overproduction of H_2_O_2_, Followed by a General Deregulation of Different Cellular Biomarkers

To be effective, herbicides must interfere with some weed physiological processes, disturb certain cellular/metabolic processes, or inhibit the biosynthesis of cellular/molecular components by blocking a specific step of a metabolic pathway [45]. Consequently, irreparable damages and tissue injuries are induced in weeds, ultimately leading to cell death and thus, their elimination [45].

The induction of oxidative stress seems to be one of the commonest key factors behind herbicide-induced metabolic impairments [46]. Often, an overproduction of ROS takes place, followed by the occurrence of oxidative damage, translated into the oxidation of the main biomolecules, from lipids and sugars, to proteins and nucleic acids [47]. Thus, the first step in dissecting the effects of DLE in weed physiology was the evaluation of the redox homeostasis in both shoots and roots of the treated plants. The obtained results showed different responses for the shoots and roots of purslane plants exposed to the DLE treatment. In the shoots, the application of DLE at 100% (v/v) significantly increased the cell H_2_O_2_ levels. In fact, ROS overproduction generally occurs in response to adverse conditions, further leading to oxidative stress, when the antioxidant (AOX) system is inhibited and/or unable to eliminate the ROS in excess [48]. In this sense, the levels of MDA, a molecule resulting from the peroxidation of membrane lipids [49], which is commonly used as an indicator of oxidative stress occurrence, were also quantified. Unexpectedly, and contrasting to the ROS levels, the shoot MDA content was not altered in any treatment. Although it may seem a little surprising, this could be the result of the registered increase in the levels of proline and carotenoids in this plant organ. Indeed, in addition to their functions as osmoprotectant and accessory photosynthetic pigment, respectively, the ability of these AOXs to scavenge ROS and to stabilize biological membranes is becoming widely recognized, now being considered potent inhibitors of LP [50,51]. Therefore, the unchanged MDA levels in shoots of purslane plants treated with DLE was most probably due to the increase in proline and carotenoid levels, along with the possible stimulation of other defense-related strategies. Accordingly, sugar and total soluble protein levels were enhanced in purslane plants exposed to DLE at 100% (v/v). Alongside proline, sugars, such as glucose and sucrose, have important AOX roles either by directly reacting with ROS, forming less harmful products, or by inducing the transcription of stress-related genes, which results in the synthesis of important AOX players [48]. In fact, the increase of the protein content observed when DLE at 100% (v/v) was applied can be indicative of the activation of gene expression, possibly resulting in the production of defense-related proteins. Indeed, it is known that under stress conditions, plants have altered gene expression, which results in the increase of the content of diverse proteins and metabolites [52]. Altogether, the obtained results highlight the occurrence of several redox disorders in shoots of DLE-exposed purslane plants, favoring the accumulation of H_2_O_2_ and the synthesis of different AOX metabolites, which prevented LP and thus, the occurrence of a severe oxidative stress condition.

With regard to the roots, the same pattern of H_2_O_2_ accumulation as in the aerial part was found. In addition, and contrary to what was observed for shoots, the increase in H_2_O_2_ levels in the roots may have exceeded the AOX system’s capacity to eliminate the excess of ROS and induced the occurrence of oxidative stress, as demonstrated by the significant increase of MDA levels at the highest concentration. Thus, it appears that the significant increase of total sugars and free amino acids in the roots of purslane plants was not enough to counteract DLE-induced oxidative stress. A redox imbalance characterized by the overproduction of ROS can produce irreversible damages in plant cells, such as the oxidation of proteins, lipids, and even nucleic acids [48]. In addition to LP, the oxidation of proteins of purslane roots might have also occurred as a consequence of the oxidative stress imposed by the application of DLE at 100% (v/v), since protein levels significantly decreased in this organ. In fact, it has been reported that the application of plant-based extracts to weeds enhances the production of the superoxide radical (O_2_^−^) and H_2_O_2_, resulting in membrane and protein damages [42]. In addition, the application of these biocides induces an electrolytic leakage that, among others, releases considerable amounts of proteases [42]. This protease release, along with the protein oxidation induced by the high concentrations of ROS, namely H_2_O_2_, might explain the great decrease in protein levels registered in *P. oleracea* roots.

In addition to oxidative stress, nutrient deficiencies or imbalances are one of the main causes that compromise plant growth [53]. Nitrogen is an essential macronutrient for plant growth and development since it is involved in the synthesis of multiple organic compounds, such as amino acids and proteins [43]. Thus, disturbances in N nutrition are responsible for significant reductions in plant growth [54]. The results of our study revealed that the foliar spraying of purslane plants with the highest concentration of DLE did not affect NO_3_^−^ assimilation in both organs, since the activity of NR was not altered. Nevertheless, a great increase in the activity of GS in the roots of *P. oleracea* plants treated with the maximum concentration of DLE was found. Glutamine is an important precursor for the biosynthesis of other amino acids and chlorophylls, among others [55]. In this sense, the increase in GS activity in the roots of the plants treated with DLE at 100% (v/v) might be responsible for the increase in the amino acid content registered in plants exposed to the same treatment, possibly in the attempt to restore the cell protein levels that were substantially affected by the oxidative burst induced by DLE application.

Overall, the obtained results pointed out that the DLE at 100% (v/v) was able to induce severe phytotoxic effects in purslane plants, considerably impairing multiple weed physiological processes. Although the DLE at 100% (v/v) was applied by foliar spraying, the phytotoxic effects in the roots of purslane plants were more pronounced than in the shoots. This may have occurred because, after 20 pulverizations, the extract ended up accumulating in the soil, being absorbed by the plant’s root system. As a matter of fact, since the application of DLE at the maximum concentration induced great inhibitory effects in the growth and development of purslane plants, their root system was unable to fully develop, compromising the function of this organ. In these conditions, the allelochemicals can easily penetrate through the primary root structure [42], causing more severe damage in the physiology of this organ than in the aerial part. In this way, the effects of DLE at 100% (v/v) in the shoots may have occurred due to the serious phytotoxic effects induced in the root system by the uptake of compounds with allelopathic activity and/or the interference of the allelochemicals with the absorption of nutrients by the roots, affecting several metabolic processes [42]. Even though great herbicidal activities against purslane seedlings have been found with the application of DLE at 250 g_dry weight_ L^−1^, its efficacy and practical applicability could be maximized if combined with preventive practices, like crop rotation and soil solarization, and other weed control methods, such as tillage [56]. Indeed, as commented by Scavo et al. [57] and Scavo et al. [58], synergies between sustainable aqueous extracts and conventional herbicides can be explored to achieve novel weed control methods. Therefore, the adoption of an integrative weed management system allows the reduction of weed populations to economically, agronomically, and ecologically acceptable thresholds, while decreasing synthetic herbicide application and their subsequent negative impacts, thus contributing to a more sustainable agriculture [56].

## 4. Materials and Methods

### 4.1. Preparation of the Aqueous Extracts

Foliage from young *E. globulus* trees, recently regenerated after a wildfire, were collected in December 2019 in a forest area of Porto, Portugal (41.197288, -8.534506) burned in the summer of 2019, according to locals. After manually detaching the leaves from the stems, a portion of the fresh leaves was oven-dried at 60 °C until reaching a constant weight. Then, fresh and dried leaves were reduced to small fragments, in order to promote the release of the allelochemicals, and underwent an extraction in deionized water at 70 °C, for 30 min, with regular agitation. The two obtained solutions were centrifuged twice (−4 °C, 15,000× *g*), for 25 and 15 min, respectively. The final concentration of the aqueous extracts corresponded to the maximum volume of leaves that could be fully immersed in 1 L of deionized water. Considering the moisture content of fresh leaves (59 ± 0.6% (w/w)), two biocides were obtained: one was prepared with dried leaves for a final concentration, in a dry weight (dw) basis, of 250 g_dw_ L^−1^ and the other was made with fresh leaves at a final concentration, in a fresh weight (fw) basis, of 617 g_fw_ L^−1^. Lastly, both aqueous extracts were filtered through nitrocellulose filters with 1.2 μm and stored at −80 °C until use. In order to functionally characterize the obtained extracts, an exhaustive analysis of their phytochemical profile was obtained by gas chromatography with mass spectrometry detection (GC-MS) and high performance liquid chromatography with UV/Vis and tandem mass spectrometry detections (HPLC-UV/Vis-MS/MS) to identify volatile and polyphenolic compounds, respectively [59]. No major differences were recorded between both extracts, but those prepared with dried leaves presented a much larger diversity of ellagitannins and greater amounts of galloyltannins, two groups of hydrolysable tannins with known allelopathic activities, according to Pinto et al. [59] and Pinto et al. [in prep].

### 4.2. Preparation of the Artificial Soil

The substrate used in all experiments consisted of a standard artificial soil, manually prepared according to the guidelines of the Organisation for Economic Co-operation and Development (OECD) [60], composed of 70% (w/w) sand, 20% (w/w) kaolin, and 10% (w/w) peat (pH (KCl) 6.0 ± 0.5).

### 4.3. Test-Species and Growth Conditions

*Portulaca oleracea* L. seeds were purchased from a local supplier and used as model weed species. Seeds were surface disinfected with 70% (v/v) ethanol for 10 min, followed by 20% (v/v) commercial bleach (5% active chlorine) for 10 min, and a series of successive washes in deionized water. Before sowing, visually damaged purslane seeds were discarded. The experiments were performed in plastic pots containing 150 g_dw_ of the artificial soil. In order to ensure the maintenance of soil moisture, a cup with deionized water was placed under each pot. At the beginning of the assays, the cups were filled with 100 mL of Hoagland solution (HS) [43], instead of deionized water, to guarantee nutrient availability. The communication between each plastic pot and the cup HS/deionized water was performed by a cotton rope inserted in a hole at the bottom of the pots. All experimental assays were performed in a growth chamber with controlled conditions (temperature: 25 °C; photoperiod: 16 h/8 h light/dark; photosynthetically active radiation (PAR): 120 μmol m^−2^ s^−1^).

### 4.4. Evaluation of the Biocidal Potential of the Aqueous Extracts in Purslane Plants—Optimization of the Most Effective Dose and Extract (FLE or DLE)

To test the herbicidal potential of the two aqueous extracts, as well as to determine the most effective formulation and concentration, FLE and DLE were foliar-applied (20 pulverizations each using a regular spray bottle, which corresponded to 15 mL of extract per pot) twice a week at different concentrations (12.5, 25, 50, 75, and 100% (v/v)), in 7-days-old purslane seedlings. The CTL seedlings were sprayed, at the same rate, with deionized water. Glyphosate (RoundUp^TM^ UltraMax, Lisbon, Portugal; 360 g L^−1^ GLY as potassium salt) was used as a positive CTL to achieve a fair comparison between the biocides and commercially available options. Based on the manufacturer’s recommendations, a single application equivalent to 10 L ha^−1^ was performed. Considering the area of the plastic pots (0.011 m^2^), this application rate corresponded to 3.96 mg of GLY per pot/replicate. To ensure that the 15 mL of the pulverization carried the adequate amount of the active ingredient (4 mg), a solution 0.267 g L^−1^ of GLY was prepared. For each treatment, 4 experimental replicates were used, each one with 20 purslane seeds sown. The percentage of viable plants, as well as their growth and macroscopic phytotoxicity symptoms were weekly monitored throughout the assay. After 5 weeks of treatments, plants were collected, separated into shoots and roots, and used for the estimation of biometric parameters (shoot and root biometry and fresh biomass) and for histochemical assays.

#### 4.4.1. Biometric Evaluation

The shoot length of purslane plants was measured from the shoot apex to the shoot–root transition. Regarding shoot and root length, each plant from each replicate was individually measured, while shoot and root fresh biomass was obtained by measuring together all the individuals from each replicate. To eliminate the interference of individuals’ number, the shoot and root weight from each replicate was divided by the number of individuals of each replicate at the end of the assay.

#### 4.4.2. Histochemical Detection of Cell Death

The cellular viability was assessed based on the protocol of Romero-Puertas et al. [61]. For this purpose, purslane shoots (one per replicate) were immersed in a solution of 0.25% (w/v) Evans Blue for 5 h. Then, the plant material was transferred to boiling 96% (v/v) ethanol in order to remove the photosynthetic pigments. The appearance of blueish spots on the leaf surface is indicative of unviable cells (cell death). The results were photographically recorded.

### 4.5. Assessment of the Phytotoxicity Induced by the Application of the Optimized Aqueous Extract in Purslane Plants

To understand the physiological processes affected by the aqueous extract application, 7-days-old purslane seedlings were foliar sprayed with DLE at 75% (v/v) and 100% (v/v)—the two most effective concentrations of the best biocide formulation. The CTL seedlings were sprayed with deionized water. For each treatment, 8 experimental replicates were considered. Five weeks after the beginning of the treatments, purslane plants were collected, separated into roots and shoots, frozen in liquid nitrogen, and stored at −80 °C for further quantification of the levels of lipid peroxidation (LP) and H_2_O_2_ and of several biomolecules (sugars, proteins, amino acids, and photosynthetic pigments) as well as the activity of N metabolism-related enzymes (nitrate reductase and glutamine synthetase). For each determination, samples from at least three experimental replicates from each experimental group (CTL, DLE at 75% (v/v) and 100% (v/v)) were used.

#### 4.5.1. LP

The spectrophotometric quantification of LP was performed by the quantification of MDA levels, using samples of plant material homogenized in 0.1% (w/v) trichloroacetic acid (TCA) [49]. After reading the absorbance (Abs) at 532 and 600 nm (unspecific turbidity) and considering an extinction coefficient of 155 mM cm^−1^, MDA levels were calculated and the results expressed in nmol g^−1^ fw.

#### 4.5.2. H_2_O_2_

The determination of H_2_O_2_ levels was carried out based on the protocol of Alexieva et al. [62], by following a spectrophotometric method in which H_2_O_2_ reacts with potassium iodide (KI; 1 M) for 1 h. After recording the Abs at 390 nm, the results were expressed as pmol g^−1^ fw, considering an extinction coefficient of 0.28 μM^−1^ cm^−1^.

#### 4.5.3. Total Sugars, Proteins, Free Amino Acids, and Proline

The levels of total sugars were quantified according to the anthrone-based method of Irigoyen et al. [63], using frozen samples of shoots and roots. After the reaction between the extract and anthrone at high temperature conditions (100 °C; 10 min), the Abs of each sample was registered at 625 nm. Sugar levels were calculated by linear regression using a curve prepared with glucose standards and results were expressed in μg g^−1^ fw. The content of total soluble proteins was quantified according to the method of Bradford [64], using frozen plant aliquots. The protein levels were expressed in mg g^−1^ fw and were calculated through a calibration curve, obtained by using different known bovine serum albumin concentrations. The levels of total free amino acids were quantified based on a ninhydrin-based protocol, as described by Lee and Takahashi [65]. A calibration curve with solutions of known concentrations of cysteine was prepared. The results were expressed in μg g^−1^ fw. The quantification of proline was based on the spectrophotometric quantification of toluene-proline complex at 520 nm, following the method described by Bates et al. [66], using ninhydrin. The proline content was expressed as μg g^−1^ fw, after calculating its levels through linear regression with standards of proline.

#### 4.5.4. Photosynthetic Pigments

The quantification of total chlorophylls and carotenoids was performed based on Lichtenthaler [67], using frozen shoot samples homogenized in 80% (v/v) acetone and quartz sand. The concentration of chlorophyll a (Chl a), chlorophyll b (Chl b), and carotenoids (Car) were calculated according to Lichtenthaler [67] formulae and expressed in a fw basis.

#### 4.5.5. Quantification of Nitrogen Metabolism-Related Enzymes

##### Nitrate Reductase

NR extraction from frozen plant samples was carried out on ice in 50 mM HEPES-KOH buffer (pH 7.8) with 1 mM phenylmethylsulfonyl fluoride (PMSF), 10 mM magnesium chloride (MgCl_2_), and 2% (w/v) polyvinylpolypyrrolidone (PVPP), following the protocol of Kaiser and Brendle-Behnisch [68]. After a 25 min centrifugation at 4 °C and 15,000× *g,* the supernatant (SN) was used for the quantification of proteins (as described in 4.5.3.) and NR activity. The quantification of NR activity was accomplished by monitoring the degradation of NADH and results were expressed as mmol min^−1^ mg^−1^ of protein, using the extinction coefficient of NADH (6.22 mM^−1^ cm^−1^).

##### Glutamine Synthetase

GS extraction and activity were done following the method of Shapiro and Stadtman [69] by homogenizing the frozen plant material in an extraction buffer containing 25 mM Tris-HCl (pH 6.4), 10 mM MgCl_2_, 1 mM dithiothreitol (DTT), 10% (v/v) glycerol, 0.05% (w/v) Triton-X, and 1% (w/v) PVPP. After centrifugation (20 min; 15,000× *g*; 4 °C), the SNs were used to determine the protein content (as described in 4.5.3.) and GS activity. The activity of GS was evaluated by the transferase assay and was expressed as nkat mg^−1^ of protein.

### 4.6. Statistics

Every experimental condition comprised at least three experimental replicates (*n* ≥ 3) and all results were expressed as mean ± standard deviation (SD). In order to test for significant differences between DLE and FLE herbicidal potential, a two-way analysis of variance (ANOVA) was executed, with the types of extract (FLE vs. DLE) defined as fixed factors, and concentrations applied (0, 12.5, 25, 50, 75 and 100% (v/v)). Whenever significant (*p* < 0.05), Tukey’s post hoc test was used to discriminate differences between the concentrations of each extract. Concerning the results reporting the phytotoxic pathways of DLE, a one-way ANOVA was performed followed by Tukey’s post hoc test, whenever *p* ≤ 0.05. Before any analysis, data were checked for homogeneity and normality, using the tests of Brown-Forsythe and Shapiro-Wilk, respectively. When appropriate, data were transformed to meet the ANOVA assumptions. All statistical analyses were carried out using GraphPad Prism^®^ 7.0 software(GraphPad Software, San Diego, California USA).

In order to test differences through time in the percentage of viable plants, a repeated-measures’ ANOVA was performed, defining the factors “within subjects” and “between subjects” as “weeks” and “concentration of the aqueous extract”. Whenever significant differences were found (*p* ≤ 0.05), by assuming the correction of Greenhouse–Geisser, a one-way ANOVA followed by Dunnett’s post hoc test was performed for each factor. This procedure was conducted in SPSS Statistics 22 (IBM SPSS^®^, Armonk, New York, USA).

## 5. Conclusions

The foliar spraying of aqueous extracts prepared with young leaves of *E. globulus*, especially when dried was revealed to have strong biocidal properties against *P. oleracea* weeds, mainly by severely disturbing the redox homeostasis of their root system. This represents the first study to unravel the herbicidal potential of the leaf biomass from young *E. globulus* trees, recently resprouted after a wildfire, which can contribute, on one hand, to decrease fire hazard by reducing the uncontrolled dispersion of this species, and on the other, to possibly increase crop production by eliminating weeds.

In the future, we intend to explore the biocidal potential of the leaves of young *E. globulus* trees in a pre-emergent scenario, and to assess their environmental safety when applied in pre- and post-emergent contexts in non-target organisms, such as crop species.

## Figures and Tables

**Figure 1 plants-10-01159-f001:**
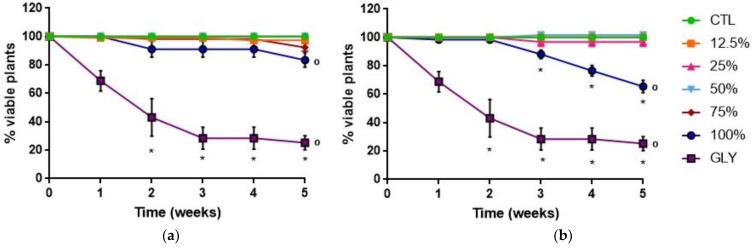
Percentage of viable plants over the five weeks of treatments with deionized water (CTL), glyphosate (GLY), and different concentrations (12.5, 25, 50, 75, and 100% (v/v)) of the aqueous extract prepared with fresh leaves (FLE; 617 g_fresh weight_ L^−1^) (**a**) and with dried leaves (DLE; 250 g_dry weight_ L^−1^) (**b**). Results are expressed as mean ± SD (*n* ≥ 3). ° refers to statistically significant differences (*p* ≤ 0.05) between treatments and the control (CTL), for the last week; * refers to statistically significant differences (*p* ≤ 0.05) between weeks, for each treatment and the CTL.

**Figure 2 plants-10-01159-f002:**
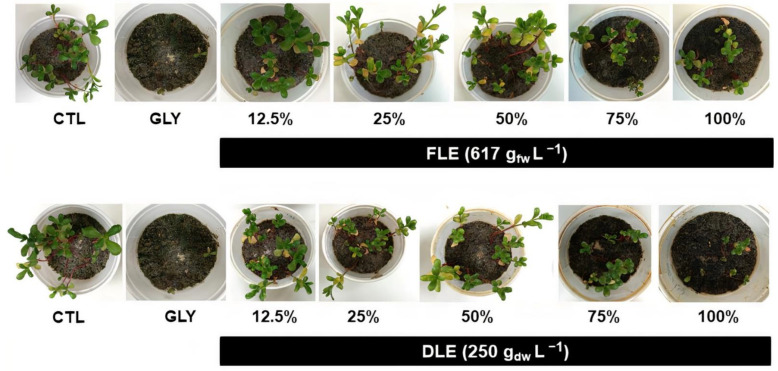
Macroscopic phytotoxicity symptoms of purslane plants after five weeks of treatments with deionized water (CTL), glyphosate (GLY), and different concentrations (12.5, 25, 50, 75, and 100% (v/v)) of the aqueous extract prepared with fresh leaves (FLE; 617 g_fresh weight_ L^−1^) and with dried leaves (DLE; 250 g_dry weight_ L^−1^).

**Figure 3 plants-10-01159-f003:**
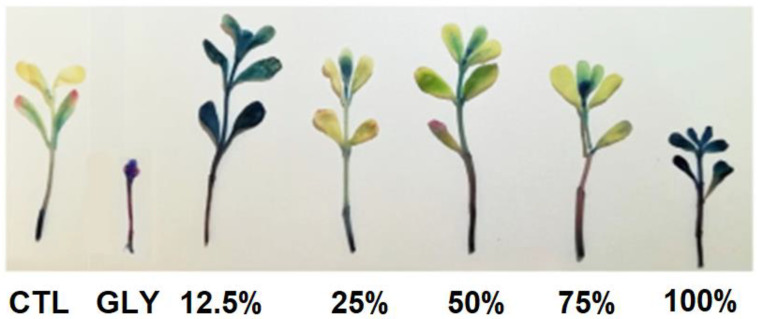
Cell death staining in shoots of purslane plants treated for five weeks with deionized water (CTL), glyphosate (GLY), and different concentrations (12.5, 25, 50, 75, and 100% (v/v)) of the aqueous extract prepared with fresh leaves (FLE; 617 g_fresh weight_ L^−1^).

**Figure 4 plants-10-01159-f004:**
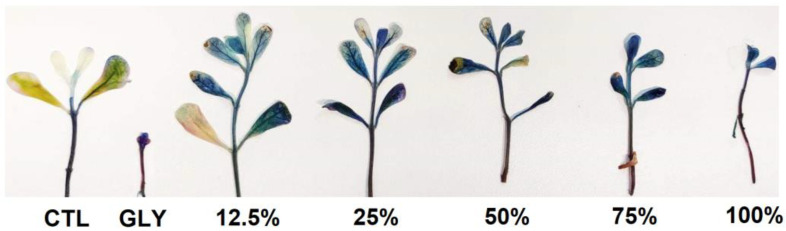
Cell death staining in shoots of purslane plants treated for five weeks with deionized water (CTL), glyphosate (GLY), and different concentrations (12.5, 25, 50, 75, and 100% (v/v)) of the aqueous extract prepared with dried leaves (DLE; 250 g_dry weight_ L^−1^).

**Table 1 plants-10-01159-t001:** Shoot and root length and fresh weight of the purslane plants treated for five weeks with deionized water (CTL), glyphosate (GLY), and different concentrations (12.5, 25, 50, 75, and 100% (v/v)) of the aqueous extract prepared with fresh leaves (FLE; 617 g_fresh weight_ L^−1^) and with dried leaves (DLE; 250 g_dry weight_ L^−1^). Results are expressed as mean ± SD (*n* ≥ 3). Different letters after the numbers indicate significant differences (*p* ≤ 0.05), according to Tukey’s post hoc test (capital letters: shoots; lowercase letters: roots).

	Endpoint	Length (cm)		Fresh Weight (g)	
Treatment		Shoots	Roots	Shoots	Roots
FLE	CTL	7.37 ± 0.30 B	6.51 ± 0.22 a	0.26 ± 0.014 C	0.020 ± 0.0034 b
	GLY	2.65 ± 0.11 D	0.79 ± 0.35 b	0.037 ± 0.0074 E	0.003 ± 0.00088 c
	12.5%	8.47 ± 0.13 A	6.13 ± 0.56 a	0.25 ± 0.026 C	0.019 ± 0.0028 b
	25%	8.85 ± 0.45 A	5.92 ± 0.78 a	0.34 ± 0.047 B	0.015 ± 0.00023 b
	50%	9.52 ± 0.44 A	7.27 ± 0.15 a	0.41 ± 0.0069 A	0.028 ± 0.0016 a
	75%	6.88 ± 0.09 B	6.43 ± 0.59 a	0.28 ± 0.0039 BC	0.029 ± 0.0011 a
	100%	5.47 ± 0.44 C	6.12 ± 0.42 a	0.18 ± 0.035 D	0.014 ± 0.0047 b
DLE	CTL	7.31 ± 0.73 B	7.11 ± 1.01 a	0.27 ± 0.024 A	0.025 ± 0.0046 ab
	GLY	2.65 ± 0.11 D	0.79 ± 0.35 c	0.037 ± 0.0074 C	0.003 ± 0.00088 d
	12.5%	7.72 ± 0.52 B	8.02 ± 0.49 a	0.25 ± 0.010 A	0.020 ± 0.0027 b
	25%	7.33 ± 0.47 B	7.20 ± 0.59 a	0.25 ± 0.043 A	0.022 ± 0.0047 ab
	50%	8.94 ± 0.77 A	6.83 ± 0.15 a	0.26 ± 0.025 A	0.027 ± 0.0031 a
	75%	6.10 ± 0.051 B	3.58 ± 0.43 b	0.13 ± 0.013 B	0.011 ± 0.0017 c
	100%	4.65 ± 0.60 C	2.46 ± 0.46 b	0.079 ± 0.017 BC	0.011 ± 0.0001 c

**Table 2 plants-10-01159-t002:** Levels of MDA, H_2_O_2_, total sugars, total free amino acids, proline, total soluble proteins, photosynthetic pigments (chlorophylls and carotenoids), and of the activity of nitrate reductase (NR) and glutamine synthetase (GS) in the shoots and roots of purslane plants treated for five weeks with deionized water (CTL) and 75% (v/v) and 100% (v/v) of the aqueous extract prepared with dried leaves (DLE; 250 g_dry weight_ L^−1^). Results are expressed as mean ± SD (*n* ≥ 3). Different letters after the numbers indicate significant differences (*p* ≤ 0.05), according to Tukey’s post hoc test (capital letters: shoots; lowercase letters: roots). “-“ denotes that the endpoint was not quantified in that organ.

	Treatment	CTL		DLE			
				75% (v/v)		100% (v/v)	
Endpoint		Shoots	Roots	Shoots	Roots	Shoots	Roots
MDA (nmol g^−1^ fw)		23.3 ± 3.4 A	3.24 ± 1.1 b	18.5 ± 3.7 A	1.53 ± 0.51 b	19.6 ± 1.1 A	6.11 ± 1.3 a
H_2_O_2_ (pmol g^−1^ fw)		13.9 ± 1.3 B	6.79 ± 0.81 a	21.8 ± 4.0 B	10.7 ± 2.5 a	43.5 ± 9.9 A	34.4 ± 3.3 b
Total sugars (μg g^−1^ fw)		1.32 ± 0.16 B	4.73 ± 2.1 b	2.00 ± 0.01 B	7.37 ± 0.61 b	3.96 ± 0.81 A	12.2 ± 1.7 a
Total free amino acids (μg g^−1^ fw)		262 ± 11 A	343 ± 36 b	195 ± 14 B	381 ± 30 b	247 ± 2.1 A	544 ± 83 a
Proline (μg g^−1^ fw)		37.1 ± 2.4 B	-	48.3 ± 6.6 AB	-	59.0 ± 15 A	-
Total soluble proteins (mg g^−1^ fw)		1.21 ± 0.14 B	1.23 ± 0.12 a	1.40 ± 0.049 B	1.27 ± 0.27 a	2.29 ± 0.35 A	0.61 ± 0.099 b
Total chlorophylls (mg g^−1^ fw)		0.315 ± 0.013 A	-	0.205 ± 0.05 B	-	0.278 ± 0.033 AB	-
Carotenoids (mg g^−1^ fw)		0.070 ± 0.001 A	-	0.066 ± 0.007 A	-	0.082 ± 0.02 B	-
NR activity (mmol min^−1^ mg^−1^ protein)		110 ± 12 B	96.4 ± 15 a	175 ± 7.3 A	111 ± 12 a	92.7 ± 7.2 B	108 ± 6.1 a
GS activity (nkat mg^−1^ protein)		4.71 ± 1.2 A	49.2 ± 7.2 b	6.88 ± 0.77 A	58.7 ± 1.6 b	7.21 ± 3.9 A	82.1 ± 6.2 a

## Data Availability

Not applicable.

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
