# Peer review of "Herbicidal Effects and Cellular Targets of Aqueous Extracts from Young Eucalyptus globulus Labill. Leaves"

_plants, 2021, doi:10.3390/plants10061159_

Round 1

Reviewer 1 Report

Dear authors,

please find my comments and suggestion in pdf document.  I think that Ms is interesting, scientific sounded and relativly well presented. However, some think need to be modified, specifically to re-move some parts from one section to onother (see in pdf). Than, I would like not to focus on post-fire management strategy since it was on the main goal of your experiment. Please omit thiis part from the title as well, and focus more on your test species -P. olearacea and generally on weed management strategy.

Author Response

We are grateful to the reviewers for the detailed evaluation of our manuscript. We have addressed their valuable comments and corrections, and all alterations have been highlighted in the text of the revised MS.

General comment: We have had in consideration the majority of the suggestions of the reviewers; when not in accordance, a justification behind the reason to disagree was given. Moreover, we have used the opportunity to double check the entire text and carried out several other minor alterations.

Reviewers’ comments (C) and authors’ answers (A)

Reviewer 1

Comments (C)

(C) Dear authors, please find my comments and suggestion in pdf document.  I think that Ms is interesting, scientific sounded and relativly well presented. However, some think need to be modified, specifically to re-move some parts from one section to onother (see in pdf). Than, I would like not to focus on post-fire management strategy since it was on the main goal of your experiment. Please omit thiis part from the title as well, and focus more on your test species -P. olearacea and generally on weed management strategy.

(A) Thank you for your evaluation. We highly appreciate the reviewers’ comments and critics. In accordance to the reviewer’s suggestions, we have made several corrections on our manuscript. Please, see our revised MS with all corrections highlighted at pink. Moreover, based on the suggestion given, we have readjusted the portion of the MS covering the importance of post-fire regenerated eucalyptus. Below, may you find some specific corrections:

L3: we agree with your perspective, so we excluded the part related to post-fire management from the title.

L33: Reference number 1 changed.

L35: Done.

L39: Done.

L49-53: Done.

L65-66: Done.

L67-68: Done.

L72: Done. Changed “pre-emergence context” to “in vitro bioassays”

L93-94: Done.

L100-107: Following the suggestions of Reviewer 1, we chose to shorten the paragraph and not exclude all information related to post-fire regeneration, since it constitutes an important premise of our work and the plant material used in the preparation of the tested extracts is the result of the resprouting process that occurs in eucalyptus trees after the occurrence of a severe disturbance, such as the occurrence of a fire. We have also included information about our model weed species, Portulaca oleracea L., as you recommended.

L126-129: Following the recommendations of Reviewer 1, we changed the introduction of the results section so that it was explicit enough without repeating ideas

L132-133: Done.

L165: Changed to a better-quality picture.

L220-222: Changed to the letter system to depict statistical differences.

L233: The appearance of blue spots in the plants treated with 12.5% (v/v) FLE is not in agreement with the other gathered results. So, we strongly believe that this outcome may have been done to some kind of experimental problem

L259: Even though the results of the biochemical determinations are all gathered in one table (the Table 2), we believe that the presence of individual titles for each group of molecules or enzymes analysed helps the reader to better locate the information.

L360-379: Rearranged that part of the discussion as it was suggested.

L397-419: Since in this paragraph we only discuss the appearance of chlorosis in the plants treated with both extracts, we think that adding here the information regarding the composition of the extracts does not seem right. So, we discussed those results in L366-370.

L414-417: We cannot delete that part of the sentence, since those results are from another study and not from our study. In that sentence, we are comparing our results with the ones obtained in that study. Yet, we included your suggestion.

L420-421: Done.

L427: Done.

L434: Done.

434-436: Done.

L441-445: Done.

L543: We included that information in the introduction (L76-86) and discussion (L339-344) sections.

L557: Done.

L582: Done.

L583-584: Done. We hope it will be more explicit this way.

L585-586: Done.

L600: Done.

L616: Done.

Reviewer 2 Report

Dear Authors,

The article submitted to the journal is written correctly. It contains all the necessary elements of experimental paper. Each of the parts is sufficiently elaborated. The only thing that suggests that the authors should pay attention to are the author guidelines. You can find minor mistakes. I have included my minor suggestions in the manuscript. Apart from that, I have no comments and I recommend the article for publication with full awareness. 

Best regards,

Reviewer

Author Response

We are grateful to the reviewers for the detailed evaluation of our manuscript. We have addressed their valuable comments and corrections, and all alterations have been highlighted in the text of the revised MS.

General comment: We have had in consideration the majority of the suggestions of the reviewers; when not in accordance, a justification behind the reason to disagree was given. Moreover, we have used the opportunity to double check the entire text and carried out several other minor alterations.

Reviewers’ comments (C) and authors’ answers (A)

Comments (C)

(C) Dear Authors,

The article submitted to the journal is written correctly. It contains all the necessary elements of experimental paper. Each of the parts is sufficiently elaborated. The only thing that suggests that the authors should pay attention to are the author guidelines. You can find minor mistakes. I have included my minor suggestions in the manuscript. Apart from that, I have no comments and I recommend the article for publication with full awareness.

(A) Thank you for your nice evaluation on our manuscript. We highly appreciate the reviewers’ comments and critics. All suggestions were incorporated in our revised MS. Please see the text highlighted at pink to check our modifications, as requested. Below, may you find some specific corrections:

L3: Done.

L74: Done.

L542, 544, 546: Done.

L557-560: Results of the chemical characterization of both extracts are being currently compiled in other manuscript, as we have conducted a major and extensive analysis of the phytochemical profile of the extracts.

L607: Done.

L639-640: Done.

L651: Done.

Reviewer 3 Report

plants-1221045 - Herbicidal effects and cellular targets of aqueous extracts from 2 young E. globulus leaves - a novel post-fire management strategy of eucalyptus stands

The manuscript entitled "Herbicidal effects and cellular targets of aqueous extracts from 2 young E. globulus leaves - a novel post-fire management strategy of eucalyptus stands", concerns laboratory experiments with the aim of evaluating the allelopathic activity of post-fire regenerated E. globulus leaves on P. oleracea growth and identifying the mechanisms of action involved.

GENERAL COMMENTS:

The subject of the manuscript falls with the general scope of the Journal and provides interesting data for the scientific community. The main concern is about the statistics, which in my opinion was not conducted properly.

In general, references are often quite outdated and, in some cases, not appropriate. Please consider a more contextualised and recent bibliography. I would like to recommend it for publication in Plants Journal, if it meets the publication policy of the Journal, after minor revision

GENERAL COMMENTS:

The subject of the manuscript falls with the general scope of the Journal and provides interesting data for the scientific community. The main concern is about the statistics, which in my opinion was not conducted properly.

In general, references are often quite outdated and, in some cases, not appropriate. Please consider a more contextualised and recent bibliography. 

SPECIFIC COMMENTS:

Abstract: it properly summarises the manuscript with the aims and main findings well explained.

  • Line 14: please add the taxonomist name
  • Line 19: please add the taxonomist name
  • Line 21: please write g DW L-1

Introduction:

  • In this Section, I suggest to add a brief mention for the use of bioherbicides with leaf aqueous extracts on the seedling growth of this cosmopolitan weed species. To improve the bibliography, I strongly reccomend adding this recent paper, as an example of use of natural substances for weed control (https://doi.org/10.4081/ija.2019.1373), in which are reported the results for sustainable alternatives to synthetic herbicides in weed control. 
  • Lines 43-45: please provide a valid definition of allelopathy first (many definitions are present in literature, for example Rice 1984 or the definition provided by the International Allelopathy Society in 1996) and then better describe the term “allelochemicals” (i.e. secondary metabolites belonging to different chemical classes with inhibitory effects on target organisms). There is a wide scientific literature on this topic.
  • Lines 65 and 66: being the second appearance, please write citriodora and E. camaldulensis.
  • Line 74: please write melongena
  • Lines 76-80: it would be useful to highlight this concept. This well-known concept in allelopathic studies indicates that the allelopathic activity of plants closely depends on donor plant’s age
  • In this research, an important factor under study is the type of leaves collected, fresh or dried. However, the authors did not report nothing about this in the background, while several researches have studied this aspect. I suggest to introduce this factor by reporting some results from the literature. I may suggest this recent manuscript in which the allelopathic potential of cultivated cardoon was analysed from fresh, dried and lyophilized leaves (https://doi.org/10.1016/j.scienta.2019.109024).

Materials and methods:

  • The methodologies were adequately explained. Unfortunately, the chemical results from the analysis of the extracts were not reported. Of course, this lack significantly lowered the scientific impact of the manuscript.
  • Statistical analysis: this paragraph needs a deep revision. Did authors verify the basic assumptions of the ANOVA prior to analysis (homoscedasticity and normality)? If yes, how? Please specify on the text
  • Statistical analysis: the type of data you reported, generally does not have a normal distribution, but the authors did not report any transformation of data. Please clarify
  • Statistical analysis: the authors used a one-way ANOVA. However, given the factors under study, they should have adopted a two-way ANOVA with “plant material” (i.e. fresh or dry) and “extract concentration” as fixed factors, in order to statistically demonstrate the effect of concentration (increase of allelopathic activity at increasing concentration) and type of leaves (increase of allelopathic activity with dry leaves). Nevertheless, it would be interesting to know the interaction of them.
  • Statistical analysis: Dunnett is a not appropriate test for mean multiple comparisons, it is too large. Please use the Least Significance Test (LSD).

Results:

  • Figure 1, Table 1 and Table 2: please add the LSD value at p<05. In the Table, in particular, it is very important given the absence of letters.
  • The quality of Figures could be improved

Discussion:

I suggest improving the discussion by adding more details about the allelopathic activity of the Eucalyptus genus, which is one of the most studied plant genus for its allelopathic potential. For instance, the allelochemicals involved in its phytotoxicity should be indicated in detail.

Moreover, in my opinion the authors should stress the most important practical application of bioherbicides, i.e. the combination with other practices. In fact, bioherbicides alone does not provide yet a sufficient weed control, while their herbicidal activity highly increases when combined with other control methods within an integrated weed management strategy. However, authors did not provide any example of practical combination examples. I strongly suggest this recent review reporting examples of combinations within IWMS (https://doi.org/10.3390/agronomy10040466), which can be very useful both in the introduction and discussion section.

Author Response

We are grateful to the reviewers for the detailed evaluation of our manuscript. We have addressed their valuable comments and corrections, and all alterations have been highlighted in the text of the revised MS.

General comment: We have had in consideration the majority of the suggestions of the reviewers; when not in accordance, a justification behind the reason to disagree was given. Moreover, we have used the opportunity to double check the entire text and carried out several other minor alterations.

Reviewers’ comments (C) and authors’ answers (A)

Reviewer 3

Comments (C)

(C) The manuscript entitled "Herbicidal effects and cellular targets of aqueous extracts from 2 young E. globulus leaves - a novel post-fire management strategy of eucalyptus stands", concerns laboratory experiments with the aim of evaluating the allelopathic activity of post-fire regenerated E. globulus leaves on P. oleracea growth and identifying the mechanisms of action involved.

GENERAL COMMENTS:

The subject of the manuscript falls with the general scope of the Journal and provides interesting data for the scientific community. The main concern is about the statistics, which in my opinion was not conducted properly.

In general, references are often quite outdated and, in some cases, not appropriate. Please consider a more contextualised and recent bibliography. I would like to recommend it for publication in Plants Journal, if it meets the publication policy of the Journal, after minor revision

(A) Thank you for your comments and critics. Based on your suggestion, we have double checked our references list and have included some new recent papers on this field. Indeed, from all the references listed, more than 66% are after 2010. Concerning the statistics, we highly appreciate your comments. Please, see the revised version of our MS, as well as our Supplementary Material with all ANOVA details.

(C) SPECIFIC COMMENTS:

Abstract: it properly summarises the manuscript with the aims and main findings well explained.

Line 14: please add the taxonomist name

Line 19: please add the taxonomist name

Line 21: please write g DW L-1

(A) All corrections were made. Changes are highlighted at pink in our revised MS.

(C) Introduction:

In this Section, I suggest to add a brief mention for the use of bioherbicides with leaf aqueous extracts on the seedling growth of this cosmopolitan weed species. To improve the bibliography, I strongly reccomend adding this recent paper, as an example of use of natural substances for weed control (https://doi.org/10.4081/ija.2019.1373), in which are reported the results for sustainable alternatives to synthetic herbicides in weed control.

Lines 43-45: please provide a valid definition of allelopathy first (many definitions are present in literature, for example Rice 1984 or the definition provided by the International Allelopathy Society in 1996) and then better describe the term “allelochemicals” (i.e. secondary metabolites belonging to different chemical classes with inhibitory effects on target organisms). There is a wide scientific literature on this topic.

(A) Thank you for your suggestions. We have revised our Introduction to meet your comments. Changes are highlighted at pink in our revised MS. Regarding the paper that you have suggested (https://doi.org/10.4081/ija.2019.1373), we decided that it would fit better in the discussion section, so we added the information there (L530-532).

(C) Lines 65 and 66: being the second appearance, please write citriodora and E. camaldulensis.

Line 74: please write melongena

Lines 76-80: it would be useful to highlight this concept. This well-known concept in allelopathic studies indicates that the allelopathic activity of plants closely depends on donor plant’s age

In this research, an important factor under study is the type of leaves collected, fresh or dried. However, the authors did not report nothing about this in the background, while several researches have studied this aspect. I suggest to introduce this factor by reporting some results from the literature. I may suggest this recent manuscript in which the allelopathic potential of cultivated cardoon was analysed from fresh, dried and lyophilized leaves (https://doi.org/10.1016/j.scienta.2019.109024).

(A) Thank you for your suggestions. We have revised our Introduction to meet your comments. Also, the manuscript was added to the references list, as this paper strengths our hypothesis that the type of material (fresh vs dried leaves) clearly influences the herbicidal potential of the extracts. Changes are highlighted at pink in our revised MS.

(C) Materials and methods:

The methodologies were adequately explained. Unfortunately, the chemical results from the analysis of the extracts were not reported. Of course, this lack significantly lowered the scientific impact of the manuscript.

Statistical analysis: this paragraph needs a deep revision. Did authors verify the basic assumptions of the ANOVA prior to analysis (homoscedasticity and normality)? If yes, how? Please specify on the text

Statistical analysis: the type of data you reported, generally does not have a normal distribution, but the authors did not report any transformation of data. Please clarify

Statistical analysis: the authors used a one-way ANOVA. However, given the factors under study, they should have adopted a two-way ANOVA with “plant material” (i.e. fresh or dry) and “extract concentration” as fixed factors, in order to statistically demonstrate the effect of concentration (increase of allelopathic activity at increasing concentration) and type of leaves (increase of allelopathic activity with dry leaves). Nevertheless, it would be interesting to know the interaction of them.

Statistical analysis: Dunnett is a not appropriate test for mean multiple comparisons, it is too large. Please use the Least Significance Test (LSD).

(A) Thank you for your suggestions. Actually, the reviewer is completely right in his/her comment on this matter. For that reason, we have reanalyzed the data concerning the herbicidal potential of both FLE and DLE. As shown in the Supplementary Material, the two-way ANOVA conducted revealed significant interactions between the type of extract (fresh vs dried) and the applied concentration (0-100%) for all tested parameters (shoot and root length and biomass production). Only after gathering this knowledge, we are able to clearly suggest that DLE was way more powerful than FLE. Furthermore, to achieve a better comparison between the mean of each treatment, Dunnet’s post-hoc test was replaced by Tukey’s one, which is similar to LSD, but corrects for multiple comparisons between groups. We have only decided to maintain Dunnet’s post-hoc in our Repeated Measures ANOVA, since that we are particularly interested in dissecting the differences between the treatments and the CTL.

(C) Results:

Figure 1, Table 1 and Table 2: please add the LSD value at p<05. In the Table, in particular, it is very important given the absence of letters.

The quality of Figures could be improved

(A) We have rearranged our tables, including the appearance of different letters to identify significant differences based on Tukey’s post-hoc test (p < 0.05). Also, figures were checked for their quality.

(C) Discussion:

I suggest improving the discussion by adding more details about the allelopathic activity of the Eucalyptus genus, which is one of the most studied plant genus for its allelopathic potential. For instance, the allelochemicals involved in its phytotoxicity should be indicated in detail.

Moreover, in my opinion the authors should stress the most important practical application of bioherbicides, i.e. the combination with other practices. In fact, bioherbicides alone does not provide yet a sufficient weed control, while their herbicidal activity highly increases when combined with other control methods within an integrated weed management strategy. However, authors did not provide any example of practical combination examples. I strongly suggest this recent review reporting examples of combinations within IWMS (https://doi.org/10.3390/agronomy10040466), which can be very useful both in the introduction and discussion section.

(A) Thank you for your evaluation. We agree with you. So, besides including the main compound classes that have been attributed to the allelopathic potential of E. globulus leaves as suggested, we have also correlated the herbicidal activities of FLE and DLE with their chemical compositions. We agree that the addition of that information can improve the manuscript quality. We think that this information would make sense to be added at the end of the discussion, as a summing up conclusion.

Round 2

Reviewer 1 Report

Dear authors,

thank you for all your corrections., the Ms is significantly improved now.   I would suggest to omit Lines 119-223 (last part of Introduction)